# Exposure to small molecule cocktails allows induction of neural crest lineage cells from human adipose-derived mesenchymal stem cells

Yuzo Takayama[1]*, Yuka Akagi[1,2], Yoichiro Shibuya[1,3], Yasuyuki S. Kida[1,2]*

1 Cellular and Molecular Biotechnology Research Institute, National Institute of Advanced Industrial Science and Technology (AIST), Tsukuba, Japan, 2 Advanced Photonics and Biosensing Open Innovation Laboratory, National Institute of Advanced Industrial Science and Technology (AIST), Tsukuba, Japan, 3 Department of Plastic and Reconstructive Surgery, University of Tsukuba, Tsukuba, Ibaraki, Japan

* yuzo-takayama@aist.go.jp (YT); y-kida@aist.go.jp (YSK)

## Abstract

Neural crest cells (NCCs) are a promising source for cell therapy and regenerative medicine owing to their multipotency, self-renewability, and capability to secrete various trophic factors. However, isolating NCCs from adult organs is challenging, because NCCs are broadly distributed throughout the body. Hence, we attempted to directly induce NCCs from human adipose-derived mesenchymal stem cells (ADSCs), which can be isolated easily, using small molecule cocktails. We established a controlled induction protocol with two-step application of small molecule cocktails for 6 days. The induction efficiency was evaluated based on mRNA and protein expression of neural crest markers, such as nerve growth factor receptor (NGFR) and sex-determining region Y-box 10 (SOX10). We also found that various trophic factors were significantly upregulated following treatment with the small molecule cocktails. Therefore, we performed global profiling of cell surface makers and identified distinctly upregulated markers, including the neural crest-specific cell surface markers CD271 and CD57. These results indicate that our chemical treatment can direct human ADSCs to developing into the neural crest lineage. This offers a promising experimental platform to study human NCCs for applications in cell therapy and regenerative medicine.

## Introduction

Neural crest cells (NCCs) are multipotent stem cells that can differentiate into peripheral neurons, Schwann cells, melanocytes, smooth muscle cells, and mesenchymal stem cells (MSCs). Owing to their multipotency, self-renewability, and ability to secrete various trophic factors which prevent cell death and maintain cellular functions, NCCs have applications in cell therapy and regenerative medicine [1, 2–4]. However, it is difficult to harvest NCCs from adult tissues because neural crest cells are broadly distributed over the whole body.

Previous studies, including our own, have reported that human iPSC-derived peripheral neurons [5, 6], melanocytes [7], and MSCs [8] are generated via the neural crest cell lineage. iPSCs possess numerous advantages such as infinite self-renewability and differentiation

**Funding:** Funding was provided by a Grant-in-Aid for Young Scientists (A) (KAKENHI: 26702015) from Japan Society for the Promotion of Science (JSPS) to YT, Grant-in-Aid for Challenging Exploratory Research (KAKENHI: 15K15649) from JSPS to YT and YSK, and Grant-in-Aid for Challenging Exploratory Research (KAKENHI: 18K19414) from JSPS to YSK. Funding was also provided by BioMimetics Sympathies Inc. to YSK. The funders had no role in study design, data collection and analysis, decision to publish, or preparation of the manuscript.

**Competing interests:** I have read the journal's policy and the authors of this manuscript have the following competing interests: YSK was funded by BioMimetics Sympathies Inc. This does not alter the authors' adherence to PLOS ONE policies on sharing data and materials.

**Abbreviations:** ADSC, Adipose-Derived Mesenchymal stem cell; BDNF, Brain Derived Neurotrophic Factor; CHIR, CHIR99021; DM, Dorsomorphin; DZNep, 3-Deazaneplanocin A; EPZ, EPZ004777; FGF2, Fibroblast Growth Factor-2; FSK, Forskolin; GDNF, Glial Cell Line-Derived Neurotrophic Factor; hTERT, human Telomerase Reverse Transcriptase; iPSC, induced Pluripotent Stem Cell; MEF2C, Myocyte Enhancer Factor-2C; MSC, Mesenchymal stem cell; NCC, Neural Crest Cel; NGF, Nerve Growth Factor; NGFR, Nerve Growth Factor Receptor; NT-3, Neurotrophin-3; PDGF-A, Platelet-Derived Growth Factor-A; SB, SB431542; SOX10, Sex-Determining Region Y-Box 10; Tranyl, Tranylcypromine; VPA, Valproic Acid.

potential, however, they also harbor the potential risks of tumorigenicity and differentiation uncertainty [9, 10]. This concern led to the research for alternative cell sources and cell induction approaches.

Viral delivery of transcriptional factors or small molecule-based signaling modulation have allowed direct conversion of somatic cells into other cell types without passing through pluripotent state [11–13]. Based on this concept, we previously reported that the combination of 7 small molecules (valproic acid (VPA), a histone deacetylase inhibitor; CHIR99021 (CHIR), a glycogen synthase kinase 3 inhibitor which activates WNT signaling; E-616452, an inhibitor of TGF-β type 1 activin-like kinase receptor (ALK) 5; Forskolin (FSK), a cAMP signaling activator; Tranylcypromine (Tranyl), a lysine-specific demethylase 1 inhibitor; Dorsomorphin (DM), a bone morphogenic protein (BMP) signaling inhibitor; SB431542 (SB), an inhibitor of TGF-β type1 ALK4, 5, 7) allowed induction of mouse embryonic fibroblasts (MEFs) into neural crest-like precursors [14]. This approach is useful for cell induction because of their cell permeability, non-immunogenicity, and ease of standardization of the small molecules. Moreover, these small molecular cocktails seemed to induce epigenetic modulations and enhancing neural crest-related signaling, and therefore induced embryonic fibroblasts into NCC lineages.

Here, we have focused on utilizing adipose-derived MSCs (ADSCs) as a potential source for NCCs. Human ADSCs already have been widely investigated and applied to regenerative medicine for their immunosuppressive function and ease of harvesting [15, 16]. Using the ADSCs, we have proposed a method for induction of NCCs from human somatic cells by optimizing small molecule combinations.

## Materials and methods

### Cell culture

All procedures were performed in accordance with the guidelines of the Committee for the Ethics on Experiments with Human Derivative Samples of the National Institute of Advanced Industrial Science and Technology (AIST) (Approval Number: 2014–169), and the Clinical Research Committee at the University of Tsukuba Hospital (Approval Number: H25-144). The human iPSC experiments were also approved by the Ethics Committee of AIST.

To obtain primary human ADSCs, subcutaneous adipose tissue destined to be discarded as surplus segments at the time of closure was harvested from patients undergoing reconstructive procedures at the University of Tsukuba Hospital after obtaining written informed consent from each donor prior to the surgeries. Harvested subcutaneous adipose tissues (~1 g) were digested with 0.4 U/ml collagenase NB (SERVA) in Hanks' Balanced Salt solution (HBSS; Thermo Fisher Scientific) containing 10% bovine serum albumin (FUJIFILM Wako Pure Chemical Industries), 1% D-(+)-glucose (Sigma-Aldrich), and 1% adenosine (Sigma-Aldrich) for 30 min at 37˚C. Dissociated human stromal vascular fraction (SVF) cells were cultured in MesenPro (Thermo Fisher Scientific) containing 5 ng/ml human recombinant basic fibroblast growth factor (bFGF; also known as FGF2, FUJIFILM Wako Pure Chemical Industries), and 1% Penicillin-streptomycin (FUJIFILM Wako Pure Chemical Industries).

hTERT (human Telomerase Reverse Transcriptase)-immortalized adipose derived mesenchymal stem cells (SCRC-4000 cells, ATCC) were also used in this study. SCRC-4000 cells were cultured in MesenPro containing 5 ng/ml human recombinant bFGF and 1% penicillin-streptomycin.

Induction of neural crest cells from human iPSCs was performed as previously reported [5]. Briefly, 201B7 human iPSCs were maintained in mTeSR1-cGMP medium (STEMCELL Technologies) on Laminin511-E8 (iMatrix511; Nippi)-coated culture plates. The culture medium was

changed daily. When the cells approached confluence, the colonies were digested into single cells using Accutase (Thermo Fisher Scientific) and the resulting cells were sub-cultured or induced as described below. Before cell induction, the collected human iPSCs were seeded onto 6-well plates (Corning) coated with 2-methacryloyloxyethyl phosphorylcholine (MPC) (Lipidure CM5206; NOF) at a density of $1\times 10^6$ cells/well. The MPC-coated culture plates were maintained on a rotary shaker at 95 rpm (OS-762RC). Cells were cultured in mTeSR1-cGMP medium containing 10 μM Y-27632 (FUJIFILM Wako Pure Chemical Industries) for 3 days to form embryoid bodies (EBs). To induce neural crest induction, EBs were transferred into and cultured in knockout serum replacement (KSR) medium containing 2 μM DM (Sigma-Aldrich), 10 μM SB (Sigma-Aldrich), and 10 ng/mL bFGF (FUJIFILM Wako Pure Chemical Industries) for 2 days. The KSR medium was comprised of DMEM-F12 (FUJIFILM Wako Pure Chemical Industries), 20% KSR (Life Technologies), 1% non-essential amino acids (FUJIFILM Wako Pure Chemical Industries), 1% monothioglycerol (FUJIFILM Wako Pure Chemical Industries), and 1% penicillin-streptomycin (FUJIFILM Wako Pure Chemical Industries). Then, the medium was changed to KSR medium containing 3 μM CHIR (Cayman Chemical) and 20 μM SB. The KSR medium containing 3 μM CHIR and 20 μM SB was changed on alternate days for 11 days.

## Chemical treatment

Human ADSCs were plated onto each well of a 96-well plate ($4 \times 10^3$ cells/well) and were cultured for 2 days (day 0–2). To induce into neural crest lineage directly, small molecule cocktails were applied to the cultured human ADSCs. Small molecule cocktails consisted of combinations of the following drugs: 2.5 μM DM, 2.5 μM SB, 1.5 μM CHIR, 0.5 μM E-616452 (Merck Millipore), 5 μM FSK (FUJIFILM Wako Pure Chemical Industries), 20 μM ISX9 (Tocris), 0.5 mM VPA (FUJIFILM Wako Pure Chemical Industries), 5 μM Tranyl (Abcam), 50 nM 3-Deazaneplanocin A (DZNep; Cayman Chemical Company), and 3 μM EPZ004777 (EPZ; Biovision). The cocktails were mixed in induction medium (IM), which consisted of a 1:1 mixture of DMEM/Ham's F-12 (FUJIFILM Wako Pure Chemical Industries) Neurobasal medium (Thermo Fisher Scientific), 1% B27 supplement minus vitamin A (Thermo Fisher Scientific), 0.5% N2 supplement (FUJIFILM Wako Pure Chemical Industries), 100 μM cAMP (Sigma-Aldrich), 20 ng/ml bFGF, and 1% penicillin-streptomycin. The samples were cultured in this media for 6 days (day 2–8). Briefly, small molecular cocktails were applied to the samples in a two-step manner except for the cp5 method (see Figs 1 and 2). In the first step, small molecular cocktails were applied for 4 days, and in the second step for 2 days.

To maintain and proliferate induced NCCs, they were cultured in maintenance medium containing 20 ng/ml bFGF and 20 ng/ml human recombinant epidermal growth factor (EGF, FUJIFILM Wako Pure Chemical Industries). Maintenance medium is composed of DMEM/Ham's F-12, 1% N2 supplement, 1% non-essential amino acids (FUJIFILM Wako Pure Chemical Industries), and 1% Penicillin-streptomycin.

## Immunohistochemical staining

Immunohistochemical staining analysis was performed as previously reported [5, 14]. Briefly, the samples were fixed with 3.7% formaldehyde (FUJIFILM Wako Pure Chemical Industries) and permeabilized with PBS containing 0.2% Tween-20 (FUJIFILM Wako Pure Chemical Industries). Samples were then blocked with PBS containing 4% Block Ace (DS Pharma Biomedical) and 0.2% Tween-20 and then incubated with primary antibodies (mouse anti-NGFR (1:200; Advanced Targeting System) and rabbit anti-SOX10 (1:500; Abcam)) overnight at 4˚C. Samples were then washed with PBS containing 0.2% Tween-20 thrice and then incubated with secondary antibodies (anti-mouse Alexa Fluor-488 and anti-rabbit Alexa Fluor-555;

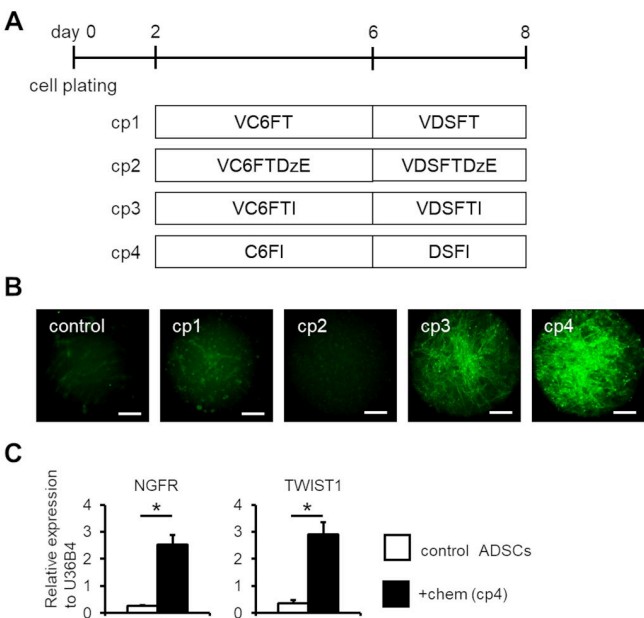

**Fig 1. Neural crest cell induction of human ADSCs.** (A) Schematic of the small molecular-based cell induction methods. Four combinations of small molecular cocktails (cp1-4) were tested. (B) Immunofluorescent analysis of NGFR expression in control and chemical-treated human ADSCs. Immunofluorescent analysis was performed on day 8 samples. Scale bar; 100 μm. (C) qPCR analysis of neural crest marker genes NGFR and TWIST1 in control and chemical (the cp4 condition)-treated human ADSCs on day 8. The expression values were normalized to that of U36B4. ($n$ = 3, error bar shows SDs; two-sided Welch's unpaired $t$-test for NGFR and two-sided Student's unpaired $t$-test for TWIST1.). V; VPA, C; CHIR, 6; E-616452, F; FSK, T; Tranyl, D; DM, S; SB, Dz; DZNep, E; EPZ, and I; ISX9. *$P < 0.01$.

1:1000; Thermo Fisher Scientific). For nuclear staining, 0.2 μg/ml Hoechst 33342 (Dojindo Molecular Technologies) was added to the samples. The samples were placed on the stage of the inverted microscope (IX81; Olympus), and fluorescence was observed with an electron multiplying charge-coupled device camera (iXon; Andor). The observed images were analyzed using ImageJ software (National Institutes of Health; available at http://imagej.nih.gov/ij/).

## RT-qPCR

RT-qPCR analysis was performed as previously reported [14]. Briefly, total RNA was isolated from the samples using the ReliaPrep RNA Cell Miniprep System (Promega). The purity and concentration of RNA were determined using a NanoDrop Lite spectrophotometer (Thermo Fisher Scientific). One hundred nanograms of total RNA were reverse-transcribed to cDNA using the ReverTra Ace qPCR RT Kit (TOYOBO, Japan). qRT-PCR was then performed using the PikoReal 96 Real-Time PCR system (Thermo Fisher Scientific) with THUNDERBIRD® SYBR qPCR Mix (TOYOBO). The expression values were normalized to U36B4 expression and are expressed as the mean ± standard deviation (SD) of triplicate measurements. The primer sequences are listed in Table 1.

## Surface antigen screening

SCRC-4000 cells ($5 \times 10^5$) were plated onto φ60 mm culture dishes and were cultured for 2 days (day 0–2). Then, cells were culture for 6 days with (chem sample) or without (control sample) small molecules (days 2–8). On day 7, cells were collected by enzymatic dissociation with

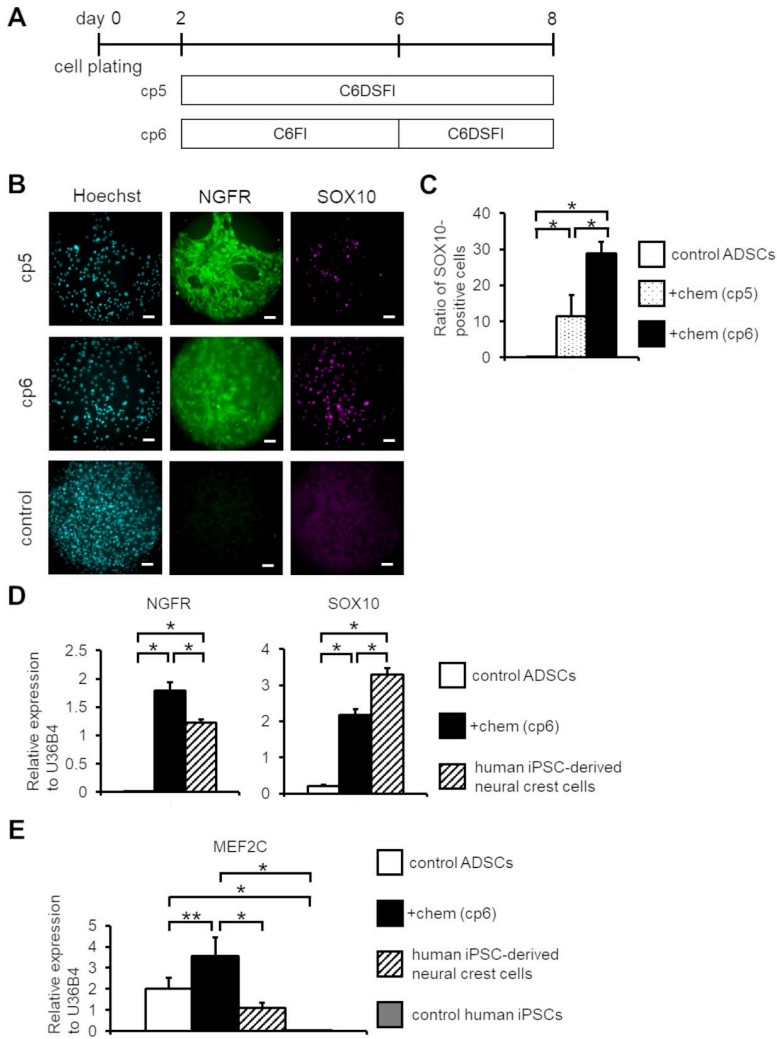

**Fig 2. Optimizing chemical induction methods to induce SOX10 expression in human ADSCs.** (A) Schematic of the cell induction methods. Two combinations of small molecular cocktails (cp5, 6) were tested on human ADSCs. (B) Immunofluorescent analysis of NGFR and SOX10 expression in control and chemical-treated human ADSCs. Immunofluorescent analysis was performed on day 8 samples. Scale bar; 100 μm. (C) Ratio of SOX10-positive cells to Hoechst-positive cells in control ADSCs and chemical (the cp5 and cp6 condition)-treated ADSCs at day 8 by immunofluorescent analysis. ($n$ = 4, error bar shows SDs; one-way ANOVA Tukey post-test). (D) qPCR analysis of NGFR and SOX10 expression among control ADSCs, chemical (the cp6 condition)-treated ADSCs on day 8, and human iPSC-derived neural crest cells. The expression values were normalized to that of U36B4. ($n$ = 3, error bar shows SDs; one-way ANOVA Tukey post-test). (E) qPCR analysis of MEF2C in control ADSCs, chemical (the cp6 condition)-treated human ADSCs on day 8, human iPSC-derived neural crest cells, and control human iPSCs. The expression values were normalized to that of U36B4. ($n$ = 3, error bar shows SDs; one-way ANOVA Tukey post-test). C; CHIR, 6; E-616452, F; FSK, D; DM, S; SB, and I; ISX9. *$P$ < 0.01, **$P$ < 0.05.

TrypLE Express and were re-plated onto 96-well plates at $1 \times 10^5$ cells/well. The next day (day 8), chem and control samples were stained with 242 antibodies from human cell surface marker screening panel (BD Lyoplate). Fluorescence was observed and analyzed as described above.

## Data analysis

Data are represented as mean + SDs (standard derivations). Differences between two groups were analyzed using Student's $t$-test. Differences among more than two groups were analyzed

**Table 1. qPCR primer sequences.**

| Gene | Forward | Reverse |
| --- | --- | --- |
| NGFR | CCTACGGCTACTACCAGGATG | CACACGGTGTTCTGCTTGT |
| TWIST1 | AAGGCATCACTATGGACTTTCTCT | GCCAGTTTGATCCCAGTATTTT |
| SOX10 | CCTCACAGATCGCCTACACC | CATATAGGAGAAGGCCGAGTAGA |
| NANOG | CCAACATCCTGAACCTCAGC | GCTATTCTTCGGCCAGTTG |
| SOX2 | TTGCTGCCTCTTTAAGACTAGGA | CTGGGGTCAAACTTCTCTC |
| MEF2C | TCGCTTGTAAATGAGGGCATACAA | GTCCAGCTTATGCCGCTGTG |
| FGF2 | TTCTTCCTGCGCATCCAC | TGCTTGAAGTTGTAGCTTGATGT |
| PDGF-A | GCAAGACCAGGACGGTCATTT | GGCACTTGACACTGCTCGT |
| BDNF | TAACGGCGGCAGACAAAAAGA | TGCACTTGGTCTCGTAGAAGTAT |
| GDNF | TTTAGGTACTGCAGCGGCTCTT | TCACTCACCAGCCTTCTATTTCTG |
| NGF | CCATCCCATCTTCCACAG | CTCTCCCAACACCATCAC |
| NT-3 | CGGAGCATAAGAGTCACC | CCTGGCTTCCTTACATCG |
| U36B4 | AGATGCAGCAGATCCGCA | GTTCTTGCCCATCAGCACC |

via one-way ANOVA, and Tukey's post hoc method was used for multiple comparisons. Differences of $P < 0.05$ were considered statistically significant.

## Results

### Screening of small molecule combinations for NCC induction

Based on our previous study, we selected the following four chemical patterns (cp1–4), and examined their effects on human ADSCs (Fig 1A); cp1: the same combination as we previously reported [14]; cp2: adding DZNep, a histone methyltransferase inhibitor [17], and EPZ, a selective inhibitor of DOT1L (a protein methyltransferase catalyzing the methylation of H3K4 [18]) to cp1; cp3: adding ISX9, inducer of neuronal differentiation [19], to cp1; and cp4: excluding VPA and Tranyl from cp3. Subsequently, NGFR expression was analyzed via immunofluorescence measurement. NGFR, also known as CD271 and p75NTR, is a transmembrane protein and a member of the tumor necrosis receptor superfamily. NGFR is a marker for early neural crest and its derivatives [20, 21]. Furthermore, knockout of NGFR gene *in vivo* caused loss of neural crest derivatives, such as peripheral neurons [22, 23]. This suggests that NGFR expression is essential for neural crest development. The expression levels of NGFR in cells induced with cp1 and cp2 conditions were comparable to control ADSCs. In contrast, the expression was distinctively upregulated in cells induced with the cp3 and cp4 conditions, particularly in the cp4 condition (Fig 1B). The results indicated that ISX9 administration was essential for induction of NGFR expression, whereas the epigenetic modulators that we utilized, such as VPA, Tranyl, DZNep, and EPZ004777, were not effective in this situation. Furthermore, we performed qPCR analysis of neural crest marker (NGFR, TWIST1) expression in ADSCs to confirm changes in mRNA expression with reference to chemical treatment. Consequently, the expression levels of both neural crest marker were significantly upregulated, by 10-fold for NGFR and 8-fold for TWIST1, in the chemical-treated human ADSCs (Fig 1C).

Although the proposed combination of small molecules (the cp4 condition) could induce NGFR-positive cells, we have found that the induced cells did not strongly express SOX10. SOX10, a member of the SOX family of transcription factors, have been identified as a central regulator in neural crest and peripheral nervous system development [24, 25]. Ectopic expression of SOX10 could convert human fibroblasts into NCCs in the presence of environmental cues [26]. These data indicate that NGFR and SOX10 are appropriate NCC markers. Therefore,

we further optimized the small molecule combinations. In accordance with the cp4 condition, we selected other combinations (cp5 and cp6), and then examined NGFR and SOX10 expression (Fig 2A). Although application of all 6 small compounds (CHIR, E-616452, DM, SB, FSK, and ISX9) together for 6 days (cp5 condition) resulted in increased expression of SOX10, this also damaged the cells. By contrast, prolonged application of CHIR and E-616452 in the cp6 condition effectively induced the expression of both NGFR and SOX10 without distinct cellular damage (Fig 2B and 2C). These results suggested that long-term actions of WNT signaling activation and TGF-β signaling inhibition are required for human NC-like cell induction. Thus, we employed the cp6 condition for induction of NCCs from human ADSCs.

It has been established that differences in human ADSCs cell lines affect their ability to proliferate and differentiate [27, 28]. Indeed, we found that proliferation rate and its response to small molecules differ based on donor variation that may be accounted for by cellular senescence status. To overcome this issue, we used hTERT immortalized-ADSCs (SCRC-4000 cells). We compared the expression levels of NGFR and SOX10 in chemical-treated immortalized-ADSCs to human iPSC-derived NCCs as a comparative control [5]. Chemical-treated immortalized-ADSCs exhibited equivalent expression level of NGFR by 1.5-fold and SOX10 by 0.7-fold compared to human iPSC-derived NCCs (Fig 2D). We also confirmed that pluripotent marker genes NANOG and SOX2 were significantly less expressed in the control and chemical-treated ADSCs than in control human iPSCs, indicating that the cp6 method did not induce cells toward pluripotency (S1 Fig). Further, we attempted to identify the involvement of myocyte enhancer factor-2C (MEF2C) expression in the cell induced with cp6. A previous report demonstrated that ISX9 administration increased the expression of MEF2 and its isoform MEF2C [19]. It is also known that MEF2C is required for proper neural crest development [29, 30]. Therefore, we performed qPCR analysis of MEF2C expressions in control ADSCs, chemical-treated (the cp6 method) ADSCs, human iPSC-derived NC cells, and control human iPSCs. We confirmed that MEF2C was not expressed in control human iPSCs, but was expressed in human iPSC-derived NC cells and chemical-treated ADSCs (Fig 2E), indicating that MEF2C is involved in NC induction. It is worth noting that MEF2C was originally expressed in control ADSCs. These data further confirm that the cp6 condition is effective for the induction of NGFR- and SOX10-positive NC-like cells from human ADSCs.

## Neuronal cytokine expression was upregulated in induced NC-like cells

Small molecules, such as FSK and ISX9, have been reported to enhance neurotrophic and neuroprotective activity [31, 32]. Thus, we explored the effects of treatment with small molecule cocktails on the trophic factor secretion capacity of induced NC-like cells. Induced NC-like cells were cultured in maintenance medium, which contained both bFGF and EGF, for 5 days after chemical treatment in order to promote complete NC induction and cell proliferation (Fig 3A) [8]. We then performed qPCR analysis for major trophic factors in control ADSCs, chemical-treated ADSCs, and human iPSC-derived NCCs (Fig 3B). Growth factor encoding genes, such as FGF2 and platelet-derived growth factor-A (PDGF-A) were concomitantly upregulated by 2- and 3-fold, respectively, in chemical-treated ADSCs compared to control ADSCs [33–35]. Additionally, neurotrophic factor encoding genes, such as BDNF (brain derived neurotrophic factor), GDNF (glial cell line-derived neurotrophic factor), NGF (nerve growth factor), and NT-3 (neurotrophin-3) were also simultaneously upregulated by 2-, 8-, 4-, and 4-fold, respectively, in chemical-treated ADSCs compared to control ADSCs. Notably, the expression levels of these trophic factors in chemical-treated ADSCs were remarkably higher than that of in human iPSC-derived NCCs. These results suggest that the small molecule cocktail treatment not only induced neural crest marker gene expression, but also enhanced expression of various trophic factors.

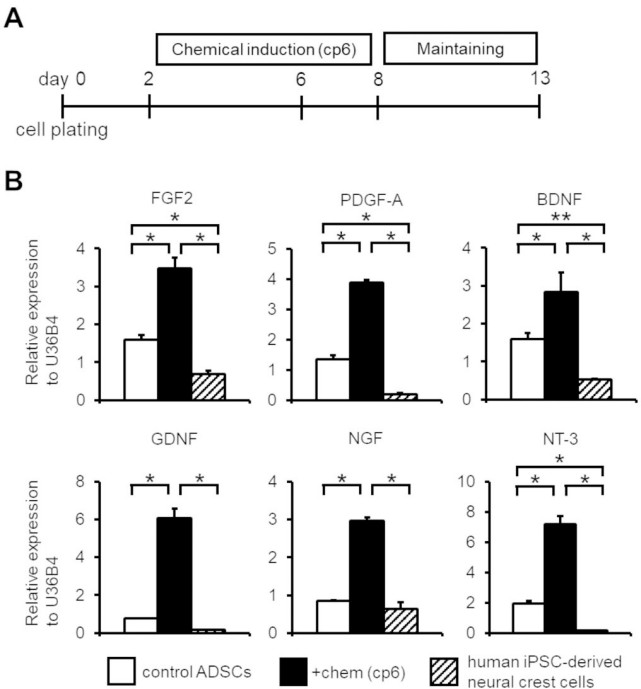

**Fig 3. Small molecular treatment enhanced expression of various trophic factor encoding genes in human ADSCs.**
(A) Schematic of the cell preparation. After the chemical induction of human ADSCs, induced cells were maintained
for 5 days in maintenance medium. (B) qPCR analysis of various trophic factor among control ADSCs, chemical (the
cp6 condition)-treated ADSCs on day 13, and human iPSC-derived neural crest cells. The expression values were
normalized to that of U36B4. (n = 3, error bar shows SDs; one-way ANOVA Tukey post-test). $^{*}P < 0.01$, $^{**}P < 0.05$.

## Profiling of induced NC-like cells based on cell surface marker expression

Detailed characterization of human ADSCs derived NC-like cells would provide further
insights into the mechanisms involved in NC induction by small molecules. To characterize
the features of induced NC-like cells, we profiled cell surface marker expression. The chemi-
cal treated immortalized-ADSCs were compared to source ADSCs as controls using 242
human CD (cluster of differentiation) antibodies provided in the BD lyoplate screening
panel (Fig 4A). We detected 6 positive markers in the source ADSCs and 21 markers in the
chemical treated immortalized-ADSCs, which have been summarized in S1 Table and S2
Fig. As expected, most source ADSC cells expressed MSC markers CD44, CD90, or CD105
[36]. In contrast, cells expressing these MSC markers were rarely observed in the chemical-
treated immortalized-ADSCs (Fig 4B). Contrarily, both of neural crest-specific cell surface
marker CD271 or CD57 (also known as BGAT1/HNK1) were expressed by chemical treated
immortalized-ADSCs (Fig 4C). Furthermore, we found that 4 cell surface markers, namely,
CD184 (also known as CXCR4), CD193 (also known as CCR3), GD2, and SSEA-4, were
strongly expressed in the induced NC-like cells (Fig 4C). CD184 and GD2 are known as neu-
ral stem cell markers [37, 38]. In addition, CD193 and SSEA-4 are well-known basophil [39]
and pluripotent stem cell markers, respectively. Their expression suggested that the induced
cells may have the other identities or potentials. Therefore, although the induced cells were
not identical to endogenous NCCs, our proposed small molecular cocktails could induce
NC-like cells from human ADSCs.

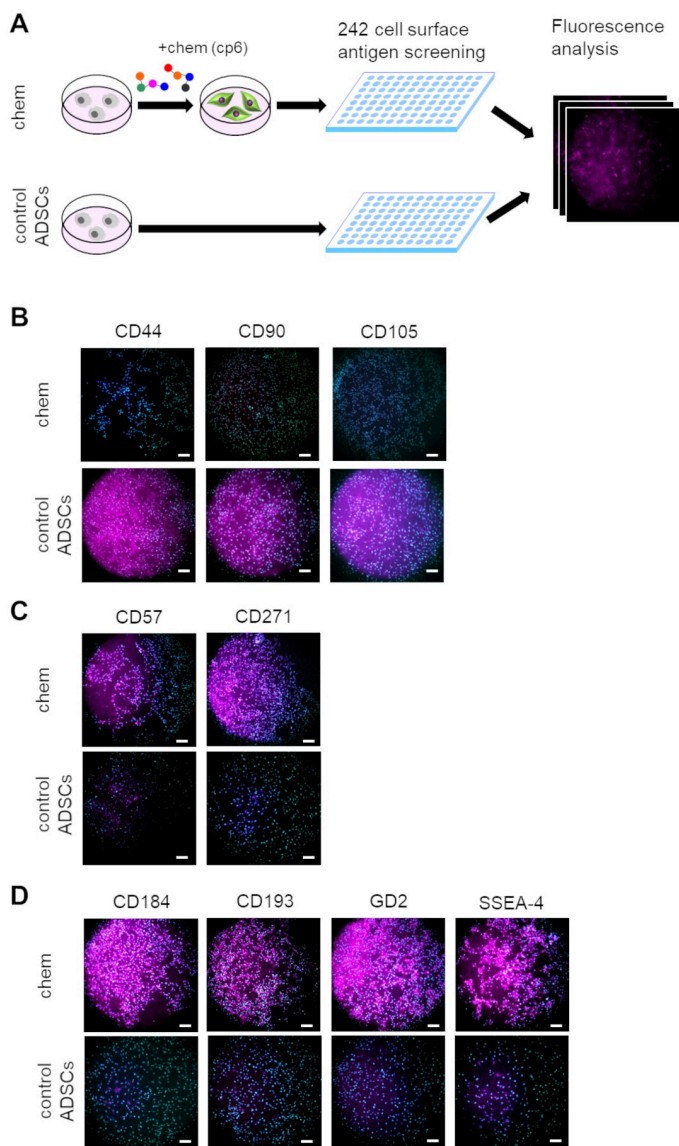

**Fig 4. Global screening of cell surface markers expressed in chemical (the cp6 condition)-treated ADSCs.** (A) Schematic of experimental flow for screening of cell surface markers. Both control and chemical treated ADSC samples were examined for global antigen screening. Expression of cell surface markers was compared based on fluorescence. (B) Immunofluorescent images of MSC markers CD44, CD90, and CD105 in control and chemical-treated ADSCs. (C) Immunofluorescent images of neural crest markers CD57 and CD271 in control and chemical-treated ADSCs. (D) Four selected cell surface markers were remarkably upregulated in chemical-treated ADSCs. Scale bar; 100 μm.

## Discussion

Human ADSCs can be harvested easily, and are transplantable somatic stem cells [4, 15, 16]. Therefore, development of methods to induce NCCs from ADSCs will be an important and versatile application of ADSCs. We previously reported that treatment with 7 small molecular compounds for 3 days allowed induction of NCCs from MEFs. In this study, found that different combinations of small molecular compounds were effective in the induction of NCCs from ADSCs.

In this study, some of the small molecules were repurposed from our previous report, indicating that similar induction pathways may be involved in ADSCs as in MEFs. Mechanistically, we considered that the combination of epigenetic modulators, such as VPA and Tranyl, reprogrammed cells to an early developmental state [40], whereas other small molecules, such as CHIR, DM, and SB, induced transdifferentiation into the NCC lineage [14]. However, we think that the ISX9 administration was particularly important, because this compound is specific to human cells. ISX9 administration has been reported to increase MEF2C expression, which is involved in NC induction [29, 30], and we confirmed that MEF2C expression was enhanced with the cp6 method. Thus, ISX9 administration may have promoted neural crest induction via MEF2C expression. However, VPA, Tranyl, DZNep, and EPZ, did not efficiently enhance induction of human ADSCs as compared to the previous study with MEFs [14]. This is due to the differences in the epigenetic status of ADSCs and embryonic fibroblasts [41, 42]. Moreover, we confirmed that MEF2C was originally expressed in control ADSCs. We speculate that the effects of ISX9 on NC induction may involve other mechanisms, along with increasing MEF2C expression.

The important characteristic of NC is its multipotency. Therefore, we performed differentiation of induced cells into peripheral neurons with neuronal differentiation medium as we previously described [5]. The induced cells showed extension of neurite-like structures after a few days of neuronal culture. However, the cells gradually died and disappeared after a one-week culture, suggesting that although the induced cells, to some extent, had the ability to differentiate into peripheral neurons, such ability was insufficient. It is possible that the reprogrammed human neurons require the supports of glial cells for their maturation [43, 44]. Therefore, further optimization of culture conditions is necessary to examine the peripheral neuron differentiation ability of induce cells.

Our chemical induction method significantly upregulated various trophic factors (Fig 3B), such as BDNF, GDNF, NGF, and NT-3, which have been reported to have therapeutic effects in neurodegenerative diseases, such as Alzheimer's disease, Parkinson's disease [45], and Amyotrophic lateral sclerosis [46]. Thus, our NC-like cells may offer a potential source of neurotrophic factors to combat neuronal damage. Further modification and analysis of the neuroprotective effects of induced NC-like cells, both *in vitro* and *in vivo*, will be required to evaluate the efficacy and usefulness of our neural crest induction strategy.

In this study, we performed global cell surface marker expression analysis to characterized induced NC-like cells (Fig 4). Distinct downregulation of MSC markers (CD44, CD90, and CD105) and upregulation of NCC markers (CD271 and CD57) across the entire cell population indicated a comprehensive induction of NC lineage cells. Furthermore, induced cells expressed CD184, CD193, GD2 or SSEA-4. CD193 is reported to be expressed in neural stem cells or neurons, and plays crucial role in neuronal injury [47, 48]. Furthermore, SSEA-4 is reported to be expressed in neural stem cells [49]. Taken together with neural stem cell-related cell surface marker CD184 and GD2, the small molecule treatment may partially induce some cells toward the (central) neural stem cell lineages. Confirming the heterogeneity of induced cells with detailed global analysis at single cell level will be required to comment on the extent of NSC induction. Moreover, profiling cell surface markers would also be useful for sorting out the specific sub-populations. Therefore, cell-isolation assays, such as FACS sorting, will provide further information regarding the identity of induced cells as well as utility of the induced NC-like cells.

## Supporting information

**S1 Table. Summary of positive CD marker expressions in control and chemical-treated immortalized-ADSCs.**
(XLSX)

**S1 Fig. qPCR analysis of pluripotent marker genes NANOG and SOX2 in control ADSCs, chemical (the cp6 condition)-treated human ADSCs on day 8, human iPSC-derived neural crest cells, and control human iPSCs.** The expression values were normalized to that of U36B4. ($n = 3$, error bar shows SDs; one-way ANOVA Tukey post-test).
(TIF)

**S2 Fig. Immunofluorescent images of CD marker expression in control and chemical-treated immortalized-ADSCs.**
(PDF)

## Acknowledgments

We thank Hiroko Kushige, Tamami Wakabayashi, Maya Iwata, Yasuko Ozaki, and Tomoko Ataka for their assistance in experimental and administrative procedures. We thank Kentaro Takagaki, Megumi Ota, and Naoki Urushihata (BioMimetics Sympathies Inc.) for support and discussion regarding this study. We would like to thank Editage (www.editage.com) for English language editing.

## Author Contributions

**Conceptualization:** Yuzo Takayama, Yasuyuki S. Kida.

**Data curation:** Yuzo Takayama, Yuka Akagi, Yasuyuki S. Kida.

**Formal analysis:** Yuka Akagi.

**Funding acquisition:** Yuzo Takayama, Yasuyuki S. Kida.

**Investigation:** Yuzo Takayama, Yasuyuki S. Kida.

**Methodology:** Yuzo Takayama, Yoichiro Shibuya.

**Project administration:** Yasuyuki S. Kida.

**Resources:** Yuzo Takayama, Yoichiro Shibuya.

**Software:** Yuzo Takayama, Yuka Akagi.

**Supervision:** Yasuyuki S. Kida.

**Validation:** Yuzo Takayama.

**Visualization:** Yuzo Takayama, Yuka Akagi.

**Writing – original draft:** Yuzo Takayama.

**Writing – review & editing:** Yuka Akagi, Yoichiro Shibuya, Yasuyuki S. Kida.

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
