## [Decision Letter · Decision Letter 0]

13 Aug 2020

PONE-D-20-22109

Exposure to small molecule cocktails allows induction of neural crest lineage cells from human adipose-derived mesenchymal stem cells

PLOS ONE

Dear Dr. Takayama,

Thank you for submitting your manuscript to PLOS ONE. After careful consideration, we feel that it has merit but does not fully meet PLOS ONE’s publication criteria as it currently stands. Therefore, we invite you to submit a revised version of the manuscript that addresses the points raised during the review process.

We look forward to receiving your revised manuscript.

Kind regards,

Michiya Matsusaki

Academic Editor

PLOS ONE

Journal Requirements:

Reviewers' comments:

Reviewer's Responses to Questions

**Comments to the Author**

1. Is the manuscript technically sound, and do the data support the conclusions?

Reviewer #1: Yes

Reviewer #2: Partly

Reviewer #3: No

2. Has the statistical analysis been performed appropriately and rigorously? 

Reviewer #1: Yes

Reviewer #2: Yes

Reviewer #3: Yes

3. Have the authors made all data underlying the findings in their manuscript fully available?

Reviewer #1: Yes

Reviewer #2: Yes

Reviewer #3: Yes

4. Is the manuscript presented in an intelligible fashion and written in standard English?

Reviewer #1: Yes

Reviewer #2: Yes

Reviewer #3: Yes

5. Review Comments to the Author

Reviewer #1: The study was designed and performed well. The text was well-written and the conclusion is important for the field. The chemically-inducible neural crest cells are potent source for cell-based therapy for neuronal damage with little variation of quality and homogeneity. However, there are several critical points to be investigated, as following.

The authors conclude the induced cells are NC-like cells. However, whether they are really functional in differentiation into neuronal lineage remains unknown. The authors should show the efficacy and potency of their NC-like cells in differentiation.

The author showed that cp5 and cp6 induce expression of SOX10 in Fig. 2B. From the image of Hoechst staining, not all cells are positive for SOX10. The author should show the percentage of double-positive cells to show the efficacy of the treatments, preferably using FACS. In addition, the author mentioned that cp5 causes cell damage and thus chose cp6, but efficacy of induction is also important. Therefore, the author should compare cp5 and cp6 in respect of the efficacy of induction of SOX10.

The NC-like cells are positive for SSEA-4 which is an established pluripotency marker, suggesting that NC-like cells induced by cp6 are reprogrammed and pretty immature. The author should stain the cells with other iPS cell marker such as SOX2 and NANOG to show which stage the NC-like cells are at.

As the authors emphasized, ISX9 is important and MEF2C might play a role in proper development of neural crest cells. Please include the data of RT-qPCR of MEF2C of the NC-like cells.

There are several minor points to be corrected

1) Stroke is not a neurodegenerative disease.

2) 616452 should be written as E-616452.

3) The relative expression by RT-qPCR are little bit confusing. Please mention relative expression to U36b4 expression in figure legends.

4) P75NTR should be p75NTR.

5) BDNF is a brain derived neurotrophic factor. Not factors. Or, do the author mean multiple family proteins? If so, provide primer sets for other members too.

6) The chemical name should be fully indicated in Introduction section. The abbreviation for tranylcypromine should be indicated there too.

Reviewer #2: 1. Fig.1C, 2C, 3B. What does the number on the vertical axis of the “Relative” expression represent? How were the data normalized?

2. In Fig.1 and Fig.2, the time course of the small molecule cocktail treatment is shown. However, the description of this time course in the main text is unclear and misleading. Please correctly describe the time course of treatment in the main text.

3. In Fig.2, human iPSC-induced NCC was used as a positive control, however, the expression of NCC marker TWIST1 was substantially low. A detailed explanation of this result would be appreciated.

4. In Fig.3, the authors have checked the expression of neurotrophic factors, which is supposed to be similar to normal NCCs. Thus, it is necessary to compare the expression level of ADSC-induced NCCs with that of normal NCCs.

5. In Fig.4, the authors have shown that ADSC-induced NCCs express cellular markers of other stem cells, which means ADSC-induced NCCs are in some degree different from normal NCCs. Therefore, whether or not ADSC-induced NCCs possess the similar multipotency of NCCs should be examined.

Reviewer #3: Author’s chemical treatment developing into the neural crest lineage is interesting. However, this study is considered to be in the process of being completed. The reliability of the condition (cp6) should be verified using different donors and 3 or more iPS strains. In addition, it is desired to induce differentiation into nerve cells. Therefore, we do not recommend publication on PLOS ONE.

---

## [Author Response · Author response to Decision Letter 0]

22 Sep 2020

Dear Editor and Reviewers,

We greatly appreciate the reviewer’s comments, which have helped us to improve the quality of our paper. The manuscript has been revised with new sentences marked in red. The new data have been incorporated into Figs. 2C, 2D, 2E, 3B, and S1 Figure. Our point-by-point responses to reviewer’s comments are below.

#Reviewer 1

The authors conclude the induced cells are NC-like cells. However, whether they are really functional in differentiation into neuronal lineage remains unknown. The authors should show the efficacy and potency of their NC-like cells in differentiation.

Our response:

We appreciate the reviewer’s informative suggestion regarding the potency of induced NC-like cells. We agree that checking NC-like potency of induced cells is important. We have performed the differentiation of induced cells into peripheral neurons with neuronal differentiation medium, which we had previously used (Takayama and Kida, PLOS ONE, 2016). The induced cells showed extension of neurite-like structures after a few days of neuronal culture. However, the cells gradually died and disappeared after a one-week culture. We speculate that the induced cells, to some extent, had the ability to differentiate into peripheral neurons; however, such ability was insufficient. This may because reprogrammed human neurons may require the support of glial cells (such as astrocytes for CNS neurons) for their maturation (Chanda et al., Stem Cell Reports, 2014; Hu et al., Cell Stem Cell, 2015). Therefore, further optimization of culture conditions would be necessary to examine the peripheral neuron differentiation ability of induce cells. However, we also consider that this is beyond the scope of this paper. We have added this description in Discussion section in the revised manuscript.

The author showed that cp5 and cp6 induce expression of SOX10 in Fig. 2B. From the image of Hoechst staining, not all cells are positive for SOX10. The author should show the percentage of double-positive cells to show the efficacy of the treatments, preferably using FACS. In addition, the author mentioned that cp5 causes cell damage and thus chose cp6, but efficacy of induction is also important. Therefore, the author should compare cp5 and cp6 in respect of the efficacy of induction of SOX10.

Our response:

We greatly appreciate the reviewer’s suggestion about the efficacy of SOX10 induction with cp5 and cp6. We have reanalyzed the immunohistochemical staining of SOX10, and confirmed that 28.9% of the induced cells expressed SOX10 with the cp6 method, in contrast to the cp5 method, which induced 11.5 % of SOX10-positive cells. Thus, we consider that the cp6 method holds promise for the induction of SOX10-positive NC-like cells.

We have added this data as Fig. 2C in the revised manuscript. We have also added this description to the first section of Result section in the revised manuscript.

The NC-like cells are positive for SSEA-4 which is an established pluripotency marker, suggesting that NC-like cells induced by cp6 are reprogrammed and pretty immature. The author should stain the cells with other iPS cell marker such as SOX2 and NANOG to show which stage the NC-like cells are at.

Our response:

We greatly appreciate the reviewer’s informative suggestion. We have performed qPCR analysis of NANOG and SOX2 expressions in control ADSC, chemical-treated (the cp6 method) SCRC, human iPSC-derived NC cells, and control human iPSCs. We confirmed that both NANOG and SOX2 were less significantly expressed in the control and chem-treated ADSCs than in control human iPSCs, indicating that the cp6 method did not induce cells toward pluripotency. Therefore, we speculate that the upregulation of SSEA-4 in the chemical-treated ADSCs indicate a partial reprogramming toward the neural stem cell lineage as described in Discussion section. We have added this data as S1 Figure in the revised manuscript. We have also added this description to first subsection of Result section in the revised manuscript

As the authors emphasized, ISX9 is important and MEF2C might play a role in proper development of neural crest cells. Please include the data of RT-qPCR of MEF2C of the NC-like cells.

Our response:

We greatly appreciate the reviewer’s informative suggestion. We agree that the confirmation of MEF2C expression after chemical treatment will be important to infer the effects of our chemical induction method. We have performed qPCR analysis of MEF2C expressions in control ADSCs (SCRC-4000 cells), chemical-treated (the cp6 method) ADSCs, human iPSC-derived NC cells, and control human iPSCs. We confirmed that MEF2C was not expressed in control human iPSCs, but was expressed in human iPSC-derived NC cells and chemical-treated ADSCs, indicating that MEF2C was involved in the induction of NC cells. Further, we confirmed that MEF2C was originally expressed in control ADSCs. We speculate that the effects of ISX9 on NC induction may involve other mechanisms as well as increasing MEF2C expression. We have added this data as Fig. 2E in the revised manuscript. We have also added this description in first subsection of Result section.

<minor points>

1) Stroke is not a neurodegenerative disease.

Our response:

We thank the reviewer for this comment. Accordingly, we have deleted the sentence describing stroke as suggested.

2) 616452 should be written as E-616452.

Our response:

We thank the reviewer for this comment. Accordingly, we have modified the sentences as suggested.

3) The relative expression by RT-qPCR are little bit confusing. Please mention relative expression to U36b4 expression in figure legends.

Our response:

We apologize for this confusion. Accordingly, we have modified the figures and figure legend as suggested.

4) P75NTR should be p75NTR.

Our response:

We thank the reviewer for this comment. Accordingly, we have modified the sentence as suggested.

5) BDNF is a brain derived neurotrophic factor. Not factors. Or, do the author mean multiple family proteins? If so, provide primer sets for other members too.

Our response:

We apologize for this confusion. We have modified the description of BDNF (brain derived neurotrophic factor) as suggested. We have also modified the description of GDNF (glial cell line-derived neurotrophic factor) and NGF (nerve growth factor).

6) The chemical name should be fully indicated in Introduction section. The abbreviation for tranylcypromine should be indicated there too.

Our response:

We apologize for this omission. Accordingly, we have modified the sentence to include the full descriptions of these abbreviations as suggested.

#Reviewer 2

Fig.1C, 2C, 3B. What does the number on the vertical axis of the “Relative” expression represent? How were the data normalized?

Our response:

We thank the reviewer for the comment and apologize for the omission. Expression level of each gene in qPCR analysis was normalized to that of the housekeeping gene U36B4 as described in the Materials and methods section. We have added this description to each figure and figure legend for clarity.

In Fig.1 and Fig.2, the time course of the small molecule cocktail treatment is shown. However, the description of this time course in the main text is unclear and misleading. Please correctly describe the time course of treatment in the main text.

Our response:

We thank the reviewer for the comment and apologize for the omission. Accordingly, we have modified the descriptions in the Material and method section as suggested.

In Fig.2, human iPSC-induced NCC was used as a positive control, however, the expression of NCC marker TWIST1 was substantially low. A detailed explanation of this result would be appreciated.

Our response:

We appreciate the reviewer’s comment and suggestion. We initially used TWIST1 gene as a NC marker; however, TWIST1 is reported as a marker for migrating NC cells (O’Rourke and Tam, Int J Dev Biol, 2002; Gammill and Bronner-Fraser, Nat Rev, 2003) and may not be suitable as a general NC marker. As the reviewer pointed out, TWIST1 is not expressed in human iPSC-derived NC cells. Therefore, we have changed the NC markers to NGFR and SOX10, and confirmed that expression levels increased to the same extent as in human iPSC-derived NC cells by chemical induction.

We have added this data as Fig. 2D. We have also modified the description in first subsection of Result section.

In Fig.3, the authors have checked the expression of neurotrophic factors, which is supposed to be similar to normal NCCs. Thus, it is necessary to compare the expression level of ADSC-induced NCCs with that of normal NCCs.

Our response:

We appreciate the reviewer’s suggestion regarding neurotrophic factor expressions. We have reanalyzed the expression of neurotrophic factors in control ADSCs (SCRC-4000 cells), chemical-treated ADSCs, and human iPSC-derived NCCs using qPCR. We confirmed that the expression of each neurotrophic factor in the chemical-treated ADSCs was significantly higher than that in human iPSC-derived NCCs. It is worth noting that as described in the Introduction, it is difficult to obtain NCCs from living organism; it is therefore difficult to use “normal NCCs” as a positive control. Thus, we have used human iPSC-derived NCCs. We have added this data as Fig. 3B. We have also modified the description in second subsection of Result section in the revised manuscript.

In Fig.4, the authors have shown that ADSC-induced NCCs express cellular markers of other stem cells, which means ADSC-induced NCCs are in some degree different from normal NCCs. Therefore, whether or not ADSC-induced NCCs possess the similar multipotency of NCCs should be examined.

Our response:

We appreciate the reviewer’s informative suggestion regarding the potency of induced NC-like cells.

We agree that checking NC-like potency of induced cells is important. We have performed differentiation of induced cells into peripheral neurons with neuronal differentiation medium, which we had previously used (Takayama and Kida, PLOS ONE, 2016). The induced cells showed extension of neurite-like structures after a few days of neuronal culture. However, the cells gradually died and disappeared after a one-week culture. Thus, we speculate that the induced cells, to some extent, had the ability to differentiate into peripheral neurons; however, such ability was insufficient. This is because reprogrammed human neurons may require the support of glial cells (such as astrocytes for CNS neurons) for their maturation (Chanda et al., Stem Cell Reports, 2014; Hu et al., Cell Stem Cell, 2015). Therefore, further optimization of culture conditions would be necessary to examine the peripheral neuron differentiation ability of induce cells. However, we also consider that this is beyond the scope of this paper. We have added this description in the Discussion section of the revised manuscript.

#Reviewer 3

We appreciate the reviewer’s honest comments. We have modified the manuscript and figures as described above based on the comments from Reviewer 1 and 2. We also would like to describe again that our chemical method can induce NC-like cells not only by using SCRC -4000 cell line but also by using ADSCs from several donors. We hope the revised paper has been improved to be worthy of the reviewer’s positive evaluation.

---

## [Decision Letter · Decision Letter 1]

9 Oct 2020

Exposure to small molecule cocktails allows induction of neural crest lineage cells from human adipose-derived mesenchymal stem cells

PONE-D-20-22109R1

Dear Dr. Takayama,

We’re pleased to inform you that your manuscript has been judged scientifically suitable for publication and will be formally accepted for publication once it meets all outstanding technical requirements.

Kind regards,

Michiya Matsusaki

Academic Editor

PLOS ONE

Additional Editor Comments (optional):

Reviewers' comments:

Reviewer's Responses to Questions

**Comments to the Author**

1. If the authors have adequately addressed your comments raised in a previous round of review and you feel that this manuscript is now acceptable for publication, you may indicate that here to bypass the “Comments to the Author” section, enter your conflict of interest statement in the “Confidential to Editor” section, and submit your "Accept" recommendation.

Reviewer #1: All comments have been addressed

Reviewer #2: All comments have been addressed

Reviewer #3: All comments have been addressed

2. Is the manuscript technically sound, and do the data support the conclusions?

Reviewer #1: Yes

Reviewer #2: Yes

Reviewer #3: Yes

3. Has the statistical analysis been performed appropriately and rigorously? 

Reviewer #1: Yes

Reviewer #2: Yes

Reviewer #3: Yes

4. Have the authors made all data underlying the findings in their manuscript fully available?

Reviewer #1: Yes

Reviewer #2: Yes

Reviewer #3: Yes

5. Is the manuscript presented in an intelligible fashion and written in standard English?

Reviewer #1: Yes

Reviewer #2: Yes

Reviewer #3: Yes

6. Review Comments to the Author

Reviewer #1: The authors responded the reviewer's comments well. Unfortunately, the NC cells died during the differentiation process, the morphological change of NC cells is convincing.

The qPCR data and immunostaining were also well performed.

Reviewer #2: The authors have properly responded to all my concerns. I also hope that this study contributes to the field of human NCCs for applications in cell therapy and regenerative medicine.

Reviewer #3: (No Response)

---

## [Editor Report · Acceptance letter]

15 Oct 2020

PONE-D-20-22109R1 

Exposure to small molecule cocktails allows induction of neural crest lineage cells from human adipose-derived mesenchymal stem cells 

Dear Dr. Takayama:

I'm pleased to inform you that your manuscript has been deemed suitable for publication in PLOS ONE. Congratulations! Your manuscript is now with our production department. 

Kind regards, 

on behalf of

Dr. Michiya Matsusaki 

Academic Editor

PLOS ONE